# A Systematic Two-Sample Mendelian Randomization Analysis Identifies Shared Genetic Origin of Endometriosis and Associated Phenotypes

**DOI:** 10.3390/life11010024

**Published:** 2021-01-03

**Authors:** Aiara Garitazelaia, Aintzane Rueda-Martínez, Rebeca Arauzo, Jokin de Miguel, Ariadna Cilleros-Portet, Sergi Marí, Jose Ramon Bilbao, Nora Fernandez-Jimenez, Iraia García-Santisteban

**Affiliations:** 1Department of Genetics, Physical Anthropology and Animal Physiology, Faculty of Medicine and Nursing, University of the Basque Country (UPV/EHU) and Biocruces-Bizkaia Health Research Institute, 48940 Leioa, Basque Country, Spain; agaritazelaia001@ikasle.ehu.eus (A.G.); arueda015@ikasle.ehu.eus (A.R.-M.); rarauzo001@ikasle.ehu.eus (R.A.); jdemiguel008@ikasle.ehu.eus (J.d.M.); ariadna_cilleros001@ehu.eus (A.C.-P.); sergi.mari@ehu.eus (S.M.); joseramon.bilbao@ehu.eus (J.R.B.); 2Spanish Biomedical Research Center in Diabetes and associated Metabolic Disorders (CIBERDEM), 28029 Madrid, Spain

**Keywords:** endometriosis, mendelian randomization, comorbidities, autoimmune diseases, anthropometric traits, female reproductive traits

## Abstract

Endometriosis, one of the most common gynecological disorders, is a complex disease characterized by the growth of endometrial-like tissue in extra-uterine locations and is a cause of pelvic pain and infertility. Evidence from observational studies indicate that endometriosis usually appears together with several other phenotypes. These include a list of autoimmune diseases, most of them more prevalent in women, anthropometric traits associated with leanness in the adulthood, as well as female reproductive traits, including altered hormone levels and those associated with a prolonged exposure to menstruation. However, the biological mechanisms underlying their co-morbidity remains unknown. To explore whether those phenotypes and endometriosis share a common genetic origin, we performed a systematic Two-Sample Mendelian Randomization (2SMR) analysis using public GWAS data. Our results suggest potential common genetic roots between endometriosis and female anthropometric and reproductive traits. Particularly, our data suggests that reduced weight and BMI might be mediating the genetic susceptibility to suffer endometriosis. Furthermore, data on female reproductive traits strongly suggest that genetic variants that predispose to a more frequent exposure to menstruation, through earlier age at menarche and shorter menstrual cycles, might also increase the risk to suffer from endometriosis.

## 1. Introduction

Endometriosis is a hormone-dependent inflammatory disorder characterized by the presence and growth of endometrial-like tissue outside the uterus, causing pelvic pain and impaired fertility [1]. Endometriosis-associated symptoms negatively impact the health and quality of life of affected women, which represent approximately 10% of reproductive-aged females, and up to 50% of women with fertility problems [2]. Besides, currently available pharmacological and surgical treatments are clearly inefficient [3].

The presentation of endometriosis is heterogeneous, and a wide variety of phenotypes appear in women with the disease [4]. In this sense, co-morbidity between endometriosis and autoimmune diseases has been consistently reported [5,6,7]. A recent meta-analysis of population-based cross-sectional, case-control, and cohort studies identified statistically significant associations between endometriosis and celiac disease, systemic lupus erythematosus, Sjögren’s syndrome, rheumatoid arthritis, multiple sclerosis and inflammatory bowel disease [8]. However, whether these immune-mediated diseases and endometriosis share common genetic origin or arise independently remains unknown.

Similarly, endometriosis has been associated with leanness and tallness in adulthood [9]. An increased odds of endometriosis has been linked to taller height [10], and inverse association between endometriosis and adult body mass index (BMI) has been reported [11,12]. In line with these observations, an opposite correlation between endometriosis and body fat distribution (measured as waist-to-hip ratio) has been described [13]. Interestingly, a genome-wide enrichment analysis identified shared loci between endometriosis and body fat distribution, but not with BMI [14], suggesting distinct mechanisms that might explain the association between each associated anthropometric trait and endometriosis.

Finally, a few, though robust risk factors frequently appear in women with endometriosis. On the one hand, hormone imbalance, mainly estrogen dominance [15] and progesterone resistance [16], are commonly observed in affected females; similarly, reduced levels of anti-Müllerian hormone (AMH) have been described in women with endometriomas [17]. On the other hand, probably as a consequence of increased exposure to menstruation, early age at menarche as well as short menstrual cycles have been associated with a greater risk of suffering from endometriosis [18,19]. Whether these associations share common genetic origins, though, remains largely unexplored.

Both endometriosis itself and the associated phenotypes are complex traits, influenced not only by environmental, but also by genetic factors. In the last decade, genome-wide association studies (GWASs) have identified single nucleotide polymorphisms (SNPs) that mediate susceptibility to suffer from those conditions. GWASs investigate complex traits using either case–control or cohort study designs, and results are usually deposited in public repositories such as the GWAS catalog or IEU GWAS database. These public GWAS results can be analyzed using Two-Sample Mendelian Randomization (2SMR), a statistical tool that enables the identification of causal and pleiotropic genetic associations between an exposure and an outcome using SNPs as instrumental variables. In the context of this work, we employed 2SMR to systematically study genetic associations between endometriosis-related phenotypes, and the disease itself. 

## 2. Materials and Methods

### 2.1. Literature Search and GWAS Data Sources

Based on the published literature, we selected a list of phenotypes that have been associated with endometriosis, and classified them into three categories: (a) Autoimmune diseases (celiac disease, systemic lupus erythematosus, Sjögren’s syndrome, rheumatoid arthritis, and multiple sclerosis), (b) Anthropometric traits (weight, height, body mass index, waist-to-hip ratio), (c) Female reproductive traits, including reproductive hormone levels (dehydroepiandrosterone sulphate, estradiol, free androgen index, follicle-stimulating hormone, luteinizing hormone, prolactin, progesterone, sex hormone-binding globulin, testosterone, anti-Müllerian hormone) and menstruation-related traits (age at menarche, age at menopause, length of menstrual cycle, and excessive and frequent menstruation).

For each of the selected traits, we conducted an extensive search in the public repositories *NHGRI-EBI GWAS catalog* (https://www.ebi.ac.uk/gwas/) and *IEU GWAS database* (https://gwas.mrcieu.ac.uk/) to select the most appropriate GWAS for each analysis (Table 1; databases were accessed in November 2020). We prioritized GWASs with large sample-sizes, cohorts of European ancestry, as well as exclusive female participation. Note that for autoimmune diseases, there was no female-only GWAS available, so we used male-female GWAS instead. For menstrual traits, we used Ben Elsworth’s UK Biobank genomic data, as these studies comprised the largest sample sizes.

### 2.2. Two-Sample Mendelian Randomization (2SMR) Analysis

Two-Sample Mendelian Randomization (2SMR) analysis was performed according to the developers’ guidelines (https://mrcieu.github.io/TwoSampleMR/index.html) using the *TwoSampleMR* R package. Figure 1 displays the analysis pipeline, detailing the commands used to perform each step.

GWASs selected to be used as exposure data (Table 1) were queried in the MR-base platform available in RStudio, and formatted using either the *format_data* (for studies from GWAS catalog) or *extract_instruments* (for studies from IEU GWAS database) functions in R. Once the instruments from the GWAS catalog were identified, the *clump_data* function was applied to ensure they were independent; instruments extracted from IEU GWAS database do not need linkage disequilibrium formatting since they have already been clumped.

Complete endometriosis GWAS summary statistics used to prepare the outcome data were obtained from the FinnGen cohort available in the IEU GWAS database using the *extract_outcome_data* function in R. The FinnGen study uses samples collected by a nationwide network of Finnish biobanks and combines genomic information with health care data including endometriosis status (https://finngen.gitbook.io/documentation/). Specifically, the GWAS with reference ID *finn-a-N14_ENDOMETRIOSIS* was used (N14 refers to diseases of the genitourinary system), comprising 3380 cases and 31,753 controls, with a total number of 16,152,119 genotyped SNPs (Table 1). 

Exposure and outcome data were subsequently harmonized using the *harmonise_data* function in R. This step is fundamental to ensure that effect and standard error on the exposure and the outcome correspond to the same effect alleles. 

The generated datasets were analyzed with the *mr* function in R using a range of methods available in the 2SMR package, including weighted median (WM), MR-Egger (MRE) and inverse variance weighted (IVW) estimates [30]. The WM method calculates Wald ratios for each of the instruments and select the median value as the causal stimate; it provides valid estimates when more than half of the SNPs satisfy the instrumental variable assumptions. The MRE method calculates Wald ratios for each of the instruments and combines the results using an adapted Egger regression; it returns an unbiased causal effect even if the assumption of no horizontal pleiotropy is violated for all SNPs. The IVW method treats each SNP as a valid natural experiment, calculates the Wald ratio for each of them and combines the results using an IVW meta-analysis approach; the slope from this approach can be interpreted as the causal effect of the exposure on the outcome [31]. 

Scatter plots were generated with the *mr_scatter_plot* function in R. Beta values were reformatted to calculate Odds Ratios (OR) and 95% Confidence Intervals (CI) to generate a forest plot for each method using the *forestplot* R package.

In addition, we obtained MR estimates of single SNPs using the *mr_singlesnp* function in R, which applies the Wald ratio method by default. Forest plots were generated with the *mr_forest_plot* function in R. The multiple-testing-adjusted *p*-value, (false discovery rate, FDR) was calculated with the Benjamini–Hochberg procedure with the function *p.adjust* of the *stats* package in R. 

## 3. Results

### 3.1. Overall 2SMR Estimates between Autoimmune Diseases, Anthropometric Traits and Female Reproductive Traits and Endometriosis

Endometriosis has been associated with multiple phenotypes, including a variety of autoimmune diseases, anthropometric traits and female reproductive traits. In order to determine whether these associations share a common genetic component, we performed a systematic 2SMR analysis (Figure 1) where GWASs for associated phenotypes were used as exposure data, and endometriosis GWAS from the FinnGen cohort as outcome data (Table 1). Figure 2 shows Mendelian Randomization estimates (Odds Ratio, 95% Confidence Interval, *p*-value) for each analysis using the weighted median (WM), MR-Egger (MRE) and inverse variance weighted (IVW) methods. The corresponding scatter plots for each analysis are represented in Appendix A
Appendix A, where effect sizes of the SNP-phenotype associations in the exposure and outcome data are plotted; in those graphs, the slopes of the lines correspond to causal estimates using each of the three different methods.

#### 3.1.1. Autoimmune Diseases and Endometriosis

Our first 2SMR analysis searched for a shared genetic component between autoimmune diseases and endometriosis. For this analysis, we selected five autoimmune diseases—celiac disease (CeD), systemic lupus erythematosus (SLE), Sjögren’s syndrome (SS), rheumatoid arthritis (RA), and multiple sclerosis (MS)—that have been previously associated with increased susceptibility to endometriosis [8]; in our selection, we prioritized those autoimmune diseases that are more prevalent in women [32]. Note that in this analysis, no female-exclusive GWAS was publicly available (Table 1). Overall 2SMR estimates show, in most of the cases, non-significant *p*-values (Figure 2) and very small and not always consistent effect-sizes (Figure 2, Appendix A
Appendix A), suggesting that the selected autoimmune diseases and endometriosis susceptibility do not share a common genetic origin. The only exception was multiple sclerosis (MS). Both MS and endometriosis are pleiotropically associated (*p* = 0.017, weighted median method), suggesting that they might share common genetic variants. According to the effect sizes reported, MS genetic susceptibility could have a protective effect for endometriosis. Of note, even though the other two methods (inverse variance weighted, MR-Egger) did not give significant associations, they showed a similar, negative correlation between effect sizes (Figure 2, Appendix A
Appendix A).

#### 3.1.2. Anthropometric Traits and Endometriosis

In our second 2SMR approach, we included four anthropometric traits—weight, height, body mass index (BMI), and Waist-To-Hip ratio (WTH)—that have been previously associated with endometriosis [9,10,11,12,13] in observational studies. Since women differ substantially from men regarding weight, height, and body fat mass and distribution (measured as means of BMI and WTH ratio, respectively), we used data from GWASs conducted separately in women for all the selected phenotypes [25,26,27] (Table 1). Scatter plots in Appendix A show that whereas SNPs related to weight and BMI display a negative correlation with endometriosis, height and WTH ratio display a positive correlation. Importantly, weight-endometriosis and BMI-endometriosis 2SMR analyses gave a significant overall pleiotropic association using the IVW (*p* = 0.015) and MR-Egger (*p* = 0.009) methods, respectively (Figure 2). Our results are in line with previous reports describing an association between adult body leanness and endometriosis and provide a putative common genetic origin of the described associations.

#### 3.1.3. Female Reproductive Traits and Endometriosis

For our last 2SMR analysis, we selected female reproductive traits, including reproductive hormone levels and menstruation-related traits, as indicated in Table 1. Genetic instruments for reproductive hormone levels were obtained from a GWAS by Ruth et al., 2016, where nine sex hormone levels [dehydroepiandrosterone sulphate (DHEAS), estradiol, free androgen index (FAI), follicle-stimulating hormone (FSH), luteinizing hormone (LH), prolactin, progesterone, sex hormone-binding globulin and testosterone] were measured in almost 3000 individuals, the majority of them (≈90%) females [28]; in addition, we included another GWAS that measured anti-Müllerian hormone (AMH) levels in pre-menopausal women [29]. Whereas the 2SMR estimates calculated with the first study were not significant, the one computed with AMH GWAS was significant in two out of the three methods employed (WM *p*-value = 0.008; IVW *p*-value = 0.001) (Figure 2). Moreover, scatter plots showed a negative slope in the three cases (Appendix A), suggesting a negative correlation between the genetic instruments of the exposure (AMH levels) and the outcome (endometriosis).

Regarding menstruation-related traits, Ben Elsworth’s UKBB data for age at menarche, age at menopause, length of menstrual cycle and excessive and frequent menstruation were selected. Note that for excessive and frequent menstruation only one SNP was selected as instrumental variable, making it impossible to calculate overall 2SMR estimates or to generate scatter plots. As shown in Figure 2, among the remaining three phenotypes, both age at menarche and length of menstrual cycle resulted in significant overall associations (*p*-value < 0.05), the latter being significant using the three methods (WM, MRE, IVW). Notably, the scatter plot for length of menstrual cycle showed a negative correlation with endometriosis (Appendix A
Appendix A), suggesting that SNPs that predispose to have longer cycles protect from endometriosis. 

### 3.2. Single SNP 2SMR Estimates of Significant Associations

For each of the selected phenotypes, we further analyzed the results at a single SNP level. Single-SNP 2SMR estimates and SNP information are summarized in Appendix A. In those cases in which overall 2SMR estimates gave significant association, we represented the results using forest plots (Figure 3). Forest plots for multiple sclerosis, weight, BMI and age at menarche show both positive and negative beta-values for individual SNPs, indicating that the SNP effects could be variably consistent or contradictory when compared to endometriosis. Interestingly, most of the SNPs predisposing to increased AMH levels and longer menstrual cycles show negative beta-values, strongly suggesting that the majority of the risk SNPs for those traits have the opposite effect in endometriosis susceptibility. Moreover, in the particular case of the phenotype “length of menstrual cycle”, global estimates gave consistently negative beta-values, with small standard errors at the single-SNP level.

Finally, we decided to further explore the individual SNPs that arose from the single SNP 2SMR analyses. In general, very few showed significant pleiotropic associations mediating endometriosis and associated traits (Appendix A). Surprisingly, three SNPs that are 25–40 Kb upstream the follicle-stimulating hormone beta subunit (FSHB) gene promoter (namely rs11031002, rs11031005 and rs11031006) appeared to be pleiotropically associated with endometriosis and four different female reproductive traits (Table 2). In particular, these traits were sex hormone levels, age at menarche, age at menopause and length of menstrual cycle. The correlation between those SNPs and endometriosis was generally negative, suggesting a different risk allele predisposing to endometriosis compared to the abovementioned female reproductive traits.

## 4. Discussion

To our knowledge, this is the first study that systematically assesses a shared genetic origin between different associated phenotypes and endometriosis using a 2SMR approach. In 2SMR studies, it is of outmost importance to carefully select the GWASs to be used as exposure and as outcome data. Genetic instruments used as exposure, which encompass a list of endometriosis-associated traits reported in literature, were extracted from public GWAS repositories by applying three selection criteria. First, since the statistical power of 2SMR analyses is determined by sample size, we prioritized genomic studies with the largest number of individuals. Second, in order to guarantee the homogeneity of the results, and to ensure the compatibility with the Finish cohort used as outcome, only GWASs with European ancestry individuals were selected. Third, given that endometriosis is a disease that affects only females, we tried to select GWASs carried out only in women, something that was possible for all the selected phenotypes except for autoimmune diseases. Regarding outcome data, since summary statistics from the largest case-control GWAS meta-analysis on endometriosis performed up to date (17,045 cases and 191,596 controls) [33] were not publicly available, we queried for public genomic data from large cohort studies. We discarded UKBB data to avoid the limitations associated with sample overlap with some of the selected exposure datasets, and we finally selected the FinnGen cohort for N14_ENDOMETRIOSIS phenotype comprising 3380 cases and 31,753 controls. 

In order to study the selected phenotypes in a systematic manner, we analyzed the overall 2SMR estimates combining all endometriosis risk SNPs. According to the literature, there is a clear co-morbidity between autoimmune diseases and endometriosis. In our study, we only found a significant association of endometriosis with MS, but our results went in the opposite direction compared to most of the published literature [6,7], as it seemed that genetic variants predisposing to one disease protect from the other. Nevertheless, this was not a robust association since we only obtained a significant *p*-value with one of the methods. In the rest of the cases, 2SMR analysis did not provide evidence of any clear causal relationship between endometriosis and the selected autoimmune diseases, and in all cases, effect sizes were close to zero. However, these data should be interpreted with caution for several reasons. First, the sample sizes for the selected GWASs are rather modest compared with the rest of the selected traits, limiting the power of the analysis. Second, exposure GWASs used for these analyses included male and female individuals, and endometriosis is a disorder only affecting women. Thus, it would be interesting to carry out a similar analysis if female autoimmune GWASs become available in the future. Even considering these caveats, our data are in line with a previous report describing the lack of genetic correlation between endometriosis and autoimmune diseases [34], suggesting that other molecular mechanisms might be mediating the link between these two types of diseases. 

The second category that we studied was a list of anthropometric traits, namely weight, height, BMI and waist-to-hip ratio, which are inversely correlated with endometriosis, according to the literature [9,10,11,12,13]. Our 2SMR analyses were in line with published reports, as the effect sizes (ORs) for most of the traits and methods were below one, suggesting a negative correlation between the exposure and outcome instrumental variables. Moreover, in the case of weight and BMI, not only the effect sizes were below one, but a significant association was also obtained in each case. These data are in line with the published literature and point to a lesser weight or BMI mediating the genetic susceptibility to suffer from endometriosis. Although due to the limited sample sizes we cannot confirm causality or mediation, in the present work we have found common genetic roots affecting predisposition to low weight or BMI and endometriosis.

The most appealing results from our 2SMR analysis derive from female reproductive traits. In the case of hormones, overall estimates of AMH levels and endometriosis yielded significant associations in two out of the three methods tested, with OR values below 1. It is well known that endometriosis may affect ovarian reserve, and therefore AMH levels, through mechanisms including ovarian injury that is caused by surgery [35]. However, our work provides a new point of view, according to which there might be a shared genetic origin between having low AMH levels and suffering from endometriosis, regardless of the AMH decrease caused by the reduction of the ovarian reserve. For all the other hormones tested, no significant associations with endometriosis were obtained. However, we have to consider that in the GWAS that we used as exposure [28], only three SNPs were significant, limiting the power of our analysis. Another possibility that might explain the lack of significant associations might be that the aberrant hormone signaling observed in women suffering from endometriosis might derive from a deregulation in hormone receptors [16,36,37], rather than from altered blood hormone levels themselves. 

For the remaining female reproductive traits analyzed, we obtained significant associations with endometriosis and ORs below one for age at menarche and length of menstrual cycle, the latter yielding significant associations with all the employed methods. Our data are in line with published literature describing increased endometriosis susceptibility with a more prolonged exposure to menstruation [18,19]. Now, our data go further, and suggest that shorter menstrual cycles could mediate the genetic susceptibility to endometriosis. Again, we cannot discard that associated SNPs provide predisposition to both shorter menstrual cycles and endometriosis without a causal link between each other, but in any case, we can confirm the existence of a common genetic basis.

Regarding particular SNPs associated to both endometriosis and different associated traits, we have found that most of the prominent associations map to a region in chromosome 11, 25–40 Kb upstream of *FSHB*. In this context, in 2016 Ruth et al. showed that a genetic variant in the same region is associated with low FSH levels and longer menstrual cycles [38]. This association was also found for both nulliparity and endometriosis risk. The authors concluded that FSH levels, genetically regulated by SNPs located upstream of the FSHB promoter, can influence a number of menstrual cycle traits such as length and menopause timing, but also reproductive outcomes, including infertility and endometriosis. Our results confirm that this genomic locus is related in a complex way to a number of female reproductive traits and affects female health outcomes and fertility.

One of the strengths of our 2SMR analysis is that it is carried out using public data. The use of public data is an effective and efficient way to generate new hypotheses, to validate the results and discoveries of previous unrelated works and to analyze existing data with modest costs in order to explore uncovered conclusions. However, we have to admit that this might also be a limitation. First, due to the limited sample sizes of existing GWASs, and to the differing numbers of cases and controls in population studies. Second, due to the fact that several of the GWASs used include males, while endometriosis studies obviously do not. Last but not least, because most of GWASs have been performed in people from European ancestry, and therefore caution should be taken when generalizing our findings to the general, non-European populations. Thus, studies like ours should be validated when more complete and powerful datasets become publicly available. In addition, larger sample sizes will probably enable to perform pleiotropy tests and will help distinguish causal associations or mediation from pleiotropic associations between unrelated events.

## 5. Conclusions

This is the first work that systematically analyzes the putative common genetic origins of endometriosis and a list of associated phenotypes using a 2SMR approach and public data. Our study serves to pinpoint common genetic roots between endometriosis and several female anthropometric and reproductive traits. Especially, we uncover a putative genetic basis of reduced AMH levels in endometriotic women, regardless of ovarian injury, and provide evidence that genetic variants that predispose to a more frequent exposure to menstruation might also increase the risk to suffer from endometriosis. Finally, we confirm the importance of the *FSHB* upstream region on different female menstrual traits and reproductive outcomes, including endometriosis disease. 

## Figures and Tables

**Figure 1 life-11-00024-f001:**
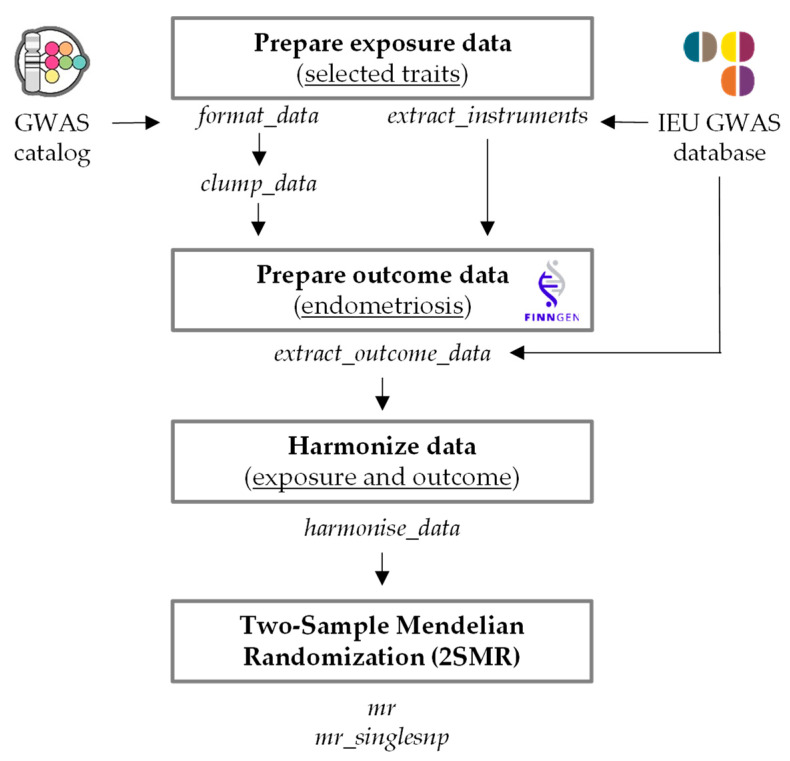
Analysis pipeline used for 2SMR. Exposure data were obtained from either GWAS catalog or IEU GWAS database and formatted and clumped when necessary. Outcome data were obtained from endometriosis FinnGen cohort and formatted to meet 2SMR requirements. Both exposure and outcome datasets were harmonized before performing the 2SMR analysis for all SNPs and for each SNP separately. Functions used in each case are indicated.

**Figure 2 life-11-00024-f002:**
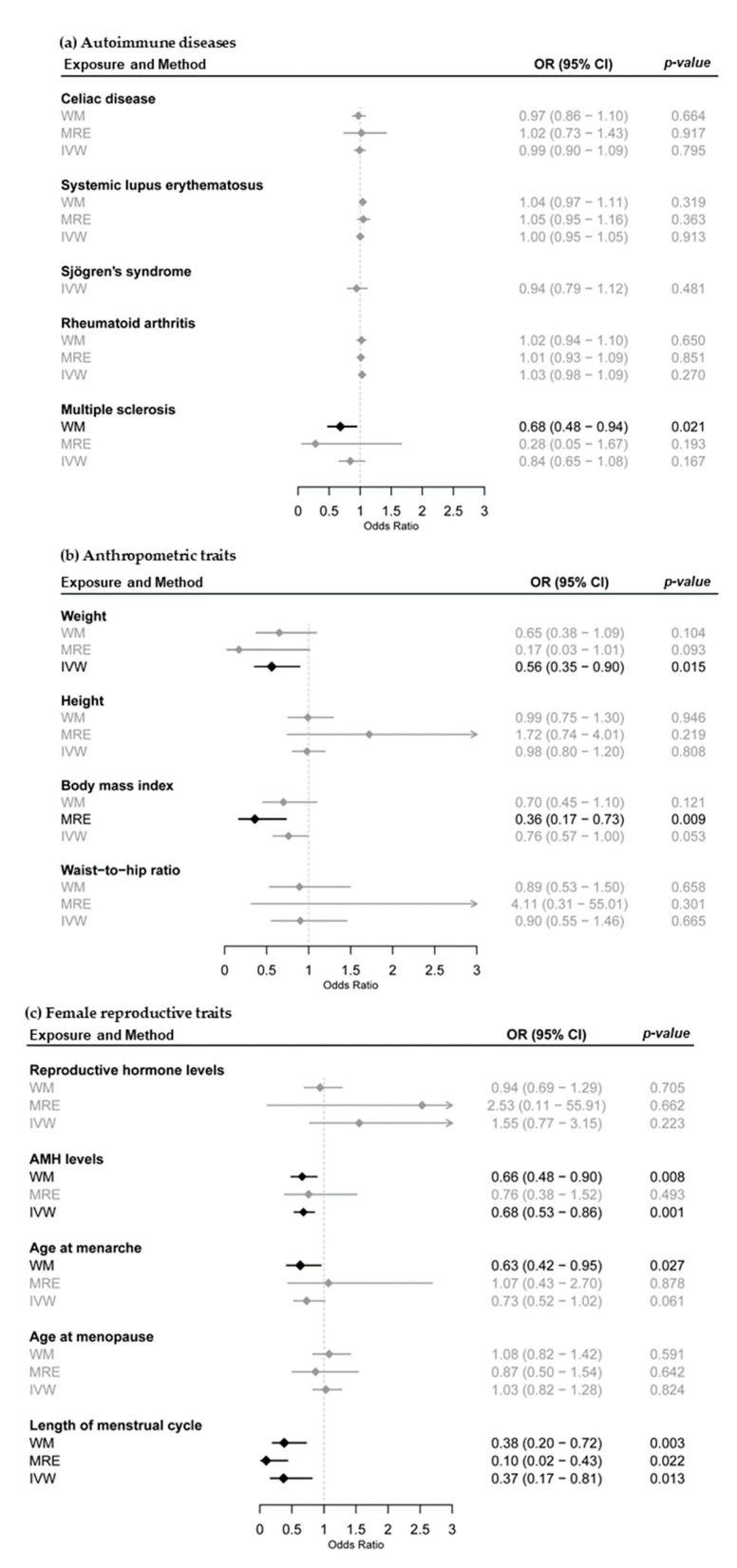
Overall 2SMR estimates between selected exposure phenotypes for (**a**) autoimmune diseases, (**b**) anthropometric traits and (**c**) female reproductive traits and endometriosis outcome. Estimates were calculated using the weighted median (WM), MR-Egger (MRE) and inverse variance weighted (IVW) methods. Effects are represented as Odds Ratio (OR) and 95% confidence interval (CI); significant associations (*p*-value < 0.05) are highlighted. AMH: Anti-Müllerian hormone.

**Figure 3 life-11-00024-f003:**
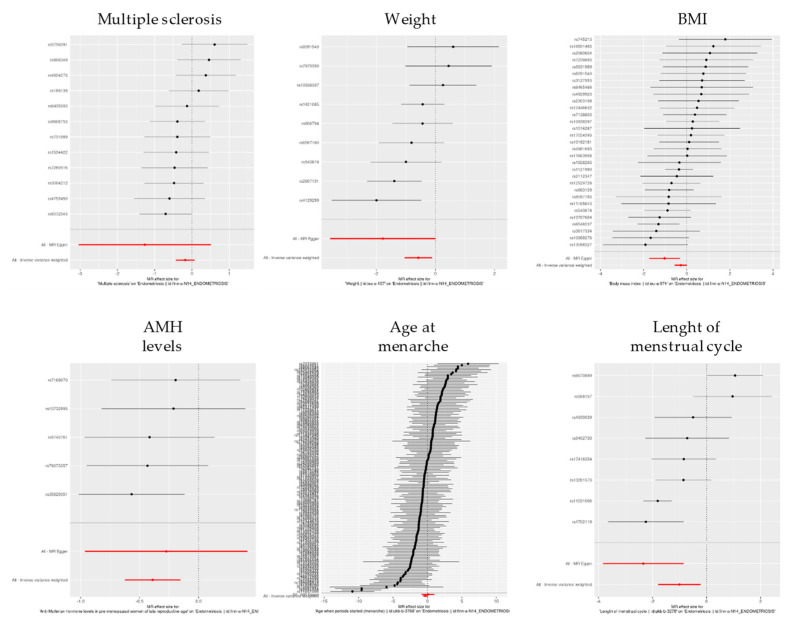
Forest plots showing individual SNP beta values (±standard error) of phenotypes that gave significant association with endometriosis in overall 2SMR analysis. BMI: Body Mass Index. AMH: Anti-Müllerian hormone.

**Table 1 life-11-00024-t001:** Phenotypes analyzed and GWASs selected as exposure/outcome data for 2SMR analysis.

Phenotype	GWAS ID	Sample Size ^1^ (Cases/Controls)	Number of SNPs	PMID [Reference]	Females Only?
**EXPOSURE DATA**					
**Autoimmune diseases**					
Celiac disease (CeD)	GCST000612	15,283	292,387	20190752 [20]	No
		(4533/10,750)			
Systemic lupus erythematosus (SLE)	GCST007400	18,264	120,873	28714469 [21]	No
		(6,748/11,516)			
Sjören’s syndrome (SS)	GCST005532	4175	648,937	24097067 [22]	No
		(1,541/2,634)			
Rheumatoid arthritis (RA)	ieu-a-832	58,284	8,747,963	24390342 [23]	No
		(14,361/43,923)			
Multiple sclerosis (MS)	GCST005531	38,589	161,311	24076602 [24]	No
		(14,498/24,091)			
**Anthropometric traits**					
Weight	ieu-a-107	73,137	2,747,007	23754948 [25]	Yes
Height	ieu-a-97	73,137	2,748,546	23754948 [25]	Yes
Body Mass Index (BMI)	ieu-a-974	171,977	2,494,613	25673413 [26]	Yes
Waist-To-Hip ratio (WTH)	ieu-a-75	118,003	2,466,102	25673412 [27]	Yes
**Female reproductive traits**					
Reproductive hormone levels ^2^	GCST002925	2913	7,879,351	26014426 [28]	Yes ^3^
AMH levels	GCST007363	3344	≈8,400,000	30649302 [29]	Yes
Age at menarche	ukb-b-3768	243,944	9,851,867	UKBB (B. Elsworth)	Yes
Age at menopause	ukb-b-17422	143,819	9,851,867	UKBB (B. Elsworth)	Yes
Length of menstrual cycle	ukb-b-3278	43,125	9,851,867	UKBB (B. Elsworth)	Yes
Excessive and frequent menstruation	ukb-b-11572	463,010	9,851,867	UKBB (B. Elsworth)	Yes
		(6,641/456,369)			
**OUTCOME DATA**					
Endometriosis	*finn-a-N14_ENDOMETRIOSIS*	35,133	16,152,119	FinnGen	Yes
		(3,380/31,753)			

^1^ In cases where the GWAS did not have a case-control design, only the sample size is displayed. ^2^ Measured reproductive hormones in GCST002925 GWAS are dehydroepiandrosterone sulphate (DHEAS), estradiol, free androgen index (FAI), follicle-stimulating hormone (FSH), luteinizing hormone (LH), prolactin, progesterone, sex hormone-binding globulin and testosterone. ^3^ From the total number of individuals in this GWAS (2913), the vast majority (2619) were women.

**Table 2 life-11-00024-t002:** Single SNP 2SMR estimates between a number of exposure phenotypes and endometriosis that share FSHB gene as the nearest TSS.

Exposure Phenotype	SNP	Beta	SE	*p*-value	FDR	Nearest TSS
Sex hormone levels	rs11031002	−1.03	0.16	2.42 × 10^−11^	1.08 × 10^−10^	FSHB
Sex hormone levels	rs11031005	0.95	0.14	2.71 × 10^−11^	1.08 × 10^−10^	FSHB
Age at menopause	rs11031005	−4.04	0.61	2.71 × 10^−11^	2.36 × 10^−9^	FSHB
Age at menarche	rs11031006	−10.91	1.63	2.42 × 10^−11^	3.82 × 10^−9^	FSHB
Length of menstrual cycle	rs11031006	−1.81	0.27	2.42 × 10^−11^	2.42 × 10^−10^	FSHB

SNP: Single Nucleotide Polymorphism; SE: Standard Error; FDR: False Discovery Rate; TSS: Transcriptional Start Site.

## Data Availability

No new data were created or analyzed in this study.

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
