# Peer review of "A Systematic Two-Sample Mendelian Randomization Analysis Identifies Shared Genetic Origin of Endometriosis and Associated Phenotypes"

_life, 2021, doi:10.3390/life11010024_

Round 1
Reviewer 1 Report
In this paper, authors described the endometriosis problem in the biological and molecular aspects. They indicated that endometriosis usually appears together with several other phenotypes. The authors used the databases (HGRI-EBI GWAS catalog(https://www.ebi.ac.uk/gwas/) and IEU GWAS database86(https://gwas.mrcieu.ac.uk/) to evaluate genetic associations between endometriosis-related phenotypes, and the disease itself.The authors were aware of the strengths of their research - as they themselves indicate, it is the first such extensive and thorough analysis, using public data, and the weaknesses of their research -limited sample sizes of existing GWASs, and the differing numbers of cases and controls in population studies, most of GWASs have been performed in people from European ancestry.As I have mentioned previously, the data is very valuable, the research was designed properly, in the logical order. Discussion included all all required items.
The paper is very interesting, well-described and designed.
I have only minor recommendations for the authors.
- Please change "&" to "and" in the whole manuscript.
- Please provide the date of access to the databases.
Reviewer 2 Report
I read with great interest the study ‘A systematic Two-Sample Mendelian Randomization analysis identifies shared genetic origin of endometriosis and associated phenotypes’.
The authors aimed to investigate the biological mechanisms underlying endometriosis and some phenotypes considered risk factors for endometriosis itself, as autoimmune diseases, anthropometric traits (leanness in adulthood), female reproductive traits (altered hormone levels, prolonged exposure to menstruation).
For the first time, they evaluate a potential common genetic origin between related phenotypes and endometriosis using a systematic Two-Sample Mendelian Randomization (2SMR) analysis and public Genome-Wide Association Studies (GWAS).
They observed that reduced weight and BMI, earlier age at menarche and shorter menstrual cycles, might increase the risk to suffer from endometriosis.
In my opinion, the study is well designed and the paper is well written. Actually, even today, endometriosis represents an enigmatic pathology and the scarce knowledge on its pathogenesis leads to a symptomatic therapy, far from the cure of its causes. The difficulties inherent to studying endometriosis are related to its multifactorial nature.
The Mendelian randomization approach, by combining genetic information into traditional epidemiological studies, might help to infer causality of risk factors on disease risk.
Investigations achieving a better understanding of physiopathological features of endometriosis constitute valuable tools to propose better tailored effective treatments.
